# Effects of *Dipsacus asperoides* and *Phlomis umbrosa* Extracts in a Rat Model of Osteoarthritis

**DOI:** 10.3390/plants10102030

**Published:** 2021-09-27

**Authors:** Jin Mi Chun, A Yeong Lee, Byeong Cheol Moon, Goya Choi, Joong-Sun Kim

**Affiliations:** 1Herbal Medicine Resources Research Center, Korea Institute of Oriental Medicine, Naju 58245, Korea; lay7709@kiom.re.kr (A.Y.L.); bcmoon@kiom.re.kr (B.C.M.); serparas@kiom.re.kr (G.C.); 2College of Veterinary Medicine, Chonnam National University, Gwangju 61186, Korea

**Keywords:** *Dipsacus asperoides*, *Phlomis umbrosa*, osteoarthritis, monosodium iodoacetate, Nagoya protocol

## Abstract

The implementation of the Nagoya Protocol highlighted the importance of identifying alternative herbal products that are as effective as traditional medicine. *Dipsacus asperoides* and *Phlomis umbrosa*, two species used in the Korean medicine ‘Sok-dan’, are used for the treatment of bone- and arthritis-related diseases, and they are often mixed or misused. To identify herbal resources with similar efficacy, we compared the effects of *D. asperoides* extract (DAE) and *P. umbrosa* extract (PUE) on osteoarthritis (OA) in a monosodium iodoacetate (MIA)-induced OA rat model. Weight-bearing distribution, serum cytokines, histopathological features, and the expression of matrix metalloproteinases (MMPs) of knee joint tissues were examined in the OA rats treated with DAE and PUE (200 mg/kg) for 21 days. DAE and PUE restored weight-bearing distribution, inhibited the production of serum cytokines, and alleviated the histopathological features of the OA knee tissue. DAE or PUE treatment decreased OA-induced overexpression of MMP-2, MMP-9, and MMP-13 in the knee joint tissue. This study demonstrated the efficacy of both DAE and PUE in an MIA-induced OA model, providing a basis for the clinical use of these products in traditional Korean medicine.

## 1. Introduction

Osteoarthritis (OA) is the most common articular disease and is a leading cause of chronic disability. The increase in the elderly population has resulted in an increase in the prevalence and risk of OA, obesity, and the rate of traumatic knee injury [1,2]. OA has therefore become a subject of much concern. The main pathological features of OA are the destruction of articular cartilage, subchondral bone sclerosis, the formation of osteophytes, inflammation of joint tissue, degeneration of the knee, and joint hypertrophy [3]. Many natural products or herbal medicines possess pharmacological properties that are suitable for the treatment of OA in experimental models [4,5].

Since the Nagoya Protocol came into force, there has been intense competition between countries to secure pre-emptive rights to state-owned indigenous resources; therefore, finding alternative herbs of similar efficacy to those used in traditional medicine is important [6]. Herbal resources are difficult to obtain, and there is increasing demand for alternative medicines of similar efficacy, which makes this an active area of research [7,8]. The identification of effective agents among Korean native herbal resources is ongoing.

*Dipsacus asperoides* and *Phlomis umbrosa* are two species used in the Korean herbal medicine ‘Sok-dan’; they are often mixed or misused alongside other agents because they have a similar name and because they are morphologically similar in dried herb form [9]. Among Korean native herbal resources, Dipsaci Radix (Sok-dan in Korean) is listed in the Korean Herbal Pharmacopoeia (KHP) as the roots of *D. asperoides* C. Y. Cheng et T. M. Ai (Caprifoliaceae family), a plant that is distributed in China for use as a medicinal herb rather than food [10]. It functions as an analgesic and anti-inflammatory agent and has been traditionally used to treat pain, rheumatic arthritis, and bone fractures [11].

Phlomidis Radix (‘Han-sok-dan’ in Korean) is also listed in the KHP as the roots of *P. umbrosa* Turczaninow (Labiatae family), a plant that is distributed in Korea and northern China and is commonly used as food and as a medicinal herb [10]. It is traditionally used for the treatment of fractures, rheumatoid arthritis, bleeding, and arthralgia [12]. *P. umbrosa* can affect bone growth [13], and it has anti-nociceptive, anti-inflammatory [14], and osteogenic effects [15]. 

Despite their difference, these agents are often mixed or misused due to their similar names in Korean herbal medicine markets and due to their morphological similarity in dried herb form [9]. For this reason, it is important to identify the authentic origin of herbal medicines, and the evaluation of the efficacy of *D. asperoides* and *P. umbrosa* is therefore necessary.

To identify alternative herbal resources of similar efficacy, the effects of *D. asperoides* and *P. umbrosa* traditional medicine need to be compared. In the present study, we compared the effects of *D. asperoides* and *P. umbrosa* on OA by evaluating weight-bearing distribution, serum cytokine production, histopathological features, and patterns of matrix metalloproteinase (MMP) expression of knee joint tissues in a monosodium iodoacetate (MIA)-induced OA rat model. 

## 2. Results

### 2.1. Effects of D. asperoides Extract (DAE) and P. umbrosa Extract (PUE) on Body Weight and Serum Markers of Toxicity

Before in vivo experiments, *D. asperoides* and *P. umbrosa* were authenticated using morphological and genetic characterization as well as phytochemical analysis for definitive authentication and quality control, as described previously [9,16,17]. In other words, to distinguish between *D. asperoides* and *P. umbrosa*, DNA barcodes were analyzed, their chloroplast genomes were sequenced, and their species were authenticated through the development of specific Sequence Characterized Amplified Region (SCAR) markers [9]. According to previous phytochemical analysis, DAE was identified to contain loganic acid, chlorogenic acid, loganin, sweroside, and isochlorogenic acid including akebia saponin D, as a specific marker for *D. asperoides.* In addition, the PUE contained sesamoside, shanzhiside methylester, and umbroside [16,17].

To assess the safety of DAE and PUE, the body weight of rats was measured for 21 days, and the results showed no significant difference in initial and final body weight between groups (Figure 1A). Serum aspartate aminotransferase (AST) and alanine aminotransferase (ALT) are enzymes linked to hepatotoxicity and are well-known biomarkers of liver damage [18], and they can be used to assess the toxicity of herbal extracts [19,20,21]. To evaluate the potential toxicity of DAE and PUE, we measured the serum markers of liver injury at sacrifice. Serum levels of AST and ALT did not differ significantly between groups (Figure 1B) and were within the normal range in rats, according to previous reports [22,23].

### 2.2. Effects of DAE and PUE on Hind Paw Weight-Bearing Distribution in MIA-Induced OA Rats

To examine the effects of DAE and PUE during OA progression, we monitored changes in hind paw weight distribution in the right (OA-induced) and left (contralateral control) limbs. The average weight-bearing distribution of the NC group was 52.6% ± 1.5%, which remained constant for 3 weeks (Figure 2). By contrast, the weight-bearing distribution of the MIA-induced control rats decreased rapidly from 7 to 21 days after MIA induction and was significantly lower than that of the NC group (*p* < 0.001). The body weight distribution between the two hind limbs was significantly lower in DAE-, PUE-, and indomethacin (IM)-treated rats than in the NC group from 7 to 21 days after MIA induction (*p* < 0.001). Compared with MIA-induced rats, DAE-treated rats showed a significant increase in weight distribution from day 7 to 14 (*p* = 0.034, *p* = 0.0008), and although there was still a slight increase on day 21, this was not statistically significant (*p* = 0.078). PUE- and IM-treated rats exhibited a significant increase in weight-bearing distribution over 21 days compared with MIA-induced rats (*p* < 0.05). These results showed that the weight distribution rate recovered gradually in both DAE- and PUE-treated groups.

### 2.3. Effects of DAE and PUE on Sezrum Levels of Cytokines in MIA-Induced OA Rats

To determine whether DAE or PUE modulate the inflammatory process by regulating the secretion of tumor necrosis factor (TNF)-α and interleukin (IL)-1β, we investigated the effects of the extracts on the serum levels of these cytokines in MIA-induced OA model rats. TNF-α and IL-1β levels were significantly higher in the MIA group than in the NC group (*p* < 0.001). By contrast, TNF-α and IL-1β levels were significantly lower in both DAE- and PUE-treated groups (*p* < 0.05, *p* < 0.01). IM treatment had similar effects (*p* < 0.01, *p* < 0.001) (Figure 3).

### 2.4. Effects of DAE and PUE on the Histopathological Features of Joint Tissues in MIA-Induced OA Rats

The histopathological features of the knee joints of rats were evaluated to assess the severity of inflammation, synovial hyperplasia, and cartilage degradation using hematoxylin and eosin (H&E) and Safranin O-fast green (Safranin O) staining. The MIA group displayed pathological changes such as joint tissue infiltration of inflammatory cells and cartilage matrix delamination compared with the NC group. Cartilage thickness in the subchondral bone was greater in DAE- and PUE-treated rats than in the MIA group. This was also observed in OA rats treated with IM as positive controls (Figure 4). These histological features indicate that treatment with DAE or PUE improved cartilage thickness and condition over those in MIA-induced rats, suggesting that the treatment decreased cartilage destruction.

### 2.5. Effects of DAE and PUE on MMP-2, MMP-9, and MMP-13 Expression in Knee Joint Tissues in MIA-Induced OA Rats

MMPs are proteolytic enzymes that regulate degradation of the extracellular matrix and are involved in OA progression [24]. The expression of MMPs involved in the degradation of the extracellular matrix in cartilage tissue was examined by immunohistochemical analysis of the activities of MMP-2, MMP-9, and MMP-13 in knee joint tissues including tibial cartilage. MMP-2 was expressed at high levels in MIA-induced rats, and expression levels were lower in DAE-, PUE-, and IM-treated rats than in the MIA treatment alone group. Similarly, the expression of MMP-9 and MMP-13 increased in MIA-induced rats and decreased in DAE-, PUE-, and IM-treated rats (Figure 5).

## 3. Discussion

This study evaluated the effects of *D. asperoides* and *P. umbrosa* on OA in an MIA-induced OA rat model to identify alternative herbal medicines with similar efficacy to traditional medicine in response to the implementation of the Nagoya protocol. Although *D. asperoides* and *P. umbrosa* belong to different species, these medicinal herbs are often mixed or misused in Korean herbal medicine. 

The origin of herbal medicines refers to ‘plants of origin’ as raw materials for correct herbal medicines in terms of authenticity, quality evaluation, and species classification of herbal medicines. Therefore, it is very important and necessary to confirm the origin of herbal medicines through various methods. In this study, the origin of these herbs was analyzed previously, and the two species were authenticated using DNA barcodes, the chloroplast genome, and SCAR markers [9]. The origin of the sample used in the present study was determined previously, and the sample was characterized by quantitative analysis [16,17]. The results showed that the major constituents of DAE included akebia saponin D, as reported previously [25]. This compound exerts anti-inflammatory and anti-nociceptive effects, as determined by different analgesic and anti-inflammatory testing methods [11]. PUE contains sesamoside, shanzhiside methylester, and umbroside, which have significant anti-nociceptive and anti-inflammatory activities [13,14]. Therefore, the origins of *D. asperoides* and *P. umbrosa* were confirmed using various authentication methods, and their major components were identified [16,17]. 

In traditional herbal medicine, *D. asperoides* is used for the treatment of pain, rheumatic arthritis, and bone fractures, whereas *P. umbrosa* is used for fractures, rheumatoid arthritis, bleeding, and arthralgia [12]. *D. asperoides* and *P. umbrosa* can affect bone growth [13], possess anti-nociceptive and anti-inflammatory activities [11,14], and have osteogenic [15] and osteoprotective [26] effects. However, the effects of *D. asperoides* and *P. umbrosa* on OA have not been examined simultaneously in a rat model of MIA-induced OA. Here, we investigated their effects on OA by evaluating weight-bearing distribution, serum cytokines productions, histopathological features, and patterns of MMPs expression in knee joint tissues in MIA-induced OA rats. 

Weight-bearing distribution is an indicator of OA progression and of the efficacy of anti-inflammatory compounds [27]. We demonstrated that DAE- and PUE-treated rats recovered hind paw weight-bearing ability compared with MIA-induced rats, and although both agents restored balance and relieved joint pain, PUE was more effective than DAE. Both agents decreased the serum levels of cytokines (TNF-α and IL-1β). Inflammation is an important factor associated with cartilage destruction in OA, and the presence of inflammatory cytokines in OA joints is related to the pathogenesis of OA. TNF-α and IL-1β play important roles in OA pathogenesis and disease severity [28]. The effects of DAE and PUE on inflammatory cytokines and their ability to elicit recovery, as demonstrated by measuring the hind paw weight-bearing distribution rate, was comparable to that of the Non-Steroidal Anti-Inflammatory Drug used as a positive control in this study. 

The analysis of the histological features showed that DAE and PUE treatment improved the cartilage thickness and condition compared with those in MIA-induced rats, which led to decreased cartilage destruction. Taken together, the results indicate that DAE and PUE alleviate OA phenotypes, including weight-bearing distribution, serum cytokine production, and histopathological features. We showed that the downregulation of MMP-2, MMP-9, and MMP-13 improved joint lesions in OA-induced rats. MMPs are a large group of enzymes responsible for matrix degradation. MMPs are expressed in the joint tissues of OA patients [29]. Among them, MMP-2 and MMP-9 are highly expressed in OA, and MMP-2 and MMP-9 activation may contribute to the cartilage destruction in OA [30]. MMP-13 is a critical target gene involved in the induction of cartilage damage during the progression of OA. MMP-13 damages articular cartilage in OA by breaking down type II collagen [31], suggesting that MMP-13 inhibition is a potential therapeutic strategy for the prevention and treatment of OA [32]. In this study, DAE or PUE treatment decreased the OA-related overexpression of MMP-2, MMP-9, and MMP-13 in the OA rat model. These results showed the indirect therapeutic effects of DAE and PUE, which were consistent with behavioral and histopathological results. 

This study provides evidence of the effects of *D. asperoides* and *P. umbrosa* on OA, indicating their potential therapeutic usefulness in OA. By evaluating both agents simultaneously, the current study provides useful information on alternative medicines with similar efficacy to known OA drugs, at least in the MIA-induced OA rat model.

Various potential natural products and herbal resources have been used to treat OA and/or to delay disease progression [4,5]. *P. umbrosa* is used as an effective functional food component [33,34], and such ethnopharmacological uses add further value to this natural resource. *P. umbrosa* is considered safe because it has been widely used in foods and traditional herbal medicines without adverse effects based on in vivo toxicological evaluation [35]. In the present study, DAE and PUE did not have toxic effects, as indicated by the normal serum AST and ALT levels, as well as the lack of significant changes in body weight in treatment groups. 

Recent reports indicate that *D. asperoides* and *P. umbrosa* exert protective effects against OA by modulating the expression of genes involved in multiple signaling pathways. *D. asperoides* treatment affects the GP6 and WNT/β-catenin signaling pathways. The effects of *P. umbrosa* on OA are mediated by its effects on the OA pathway, WNT/β-catenin, and sonic hedgehog signaling pathways [16,17]. Further studies are necessary to examine the underlying molecular mechanisms of *D. asperoides* and *P. umbrosa*, and the effects of their active compounds.

Since the Nagoya Protocol came into force, the acquisition of traditional herbal medicines has become increasingly difficult, and the demand for alternative herbs with similar efficacy has increased, which has led to comparative studies of their effects [7,8]. The domestic use of *D. asperoides* depends on imports from China, whereas *P. umbrosa* is a Korean native herb, and its availability does not depend on imports. Therefore, we compared the effects of these herbal medicines on OA and showed that the two species may be effective therapeutics for the treatment of OA.

## 4. Materials and Methods

### 4.1. Plant Materials

Dried roots of *D. asperoides* and *P. umbrosa* were purchased from Naemome Dah Herbal Medicine (Ulsan, Gyeongsangnam-do, Korea) and MyRyeung Herbal Medicine (Pocheon, Gyeonggi-do, Korea), respectively (Figure 6A). Samples used in this study were authenticated by morphological feature analysis (Dr. Goya Choi, KIOM) and by SCAR marker analysis, as described previously [9]. Voucher specimens were deposited in the Korean Herbarium of Standard Herbal Resources (No. 2-17-0059–2-17-0060 and No. 2-17-0072). 

### 4.2. Preparation of Herbal Extracts

Dried *D. asperoides* (608.9 g) and *P. umbrosa* (603.05 g) roots were refluxed in 70% ethanol for 2 h, and the extracts were filtered and evaporated in vacuo. The yields of dried *D. asperoides* and *P. umbrosa* extracts were 47.86% (*w*/*w*) and 26.62% (*w*/*w*), respectively, and samples were stored at 4 °C until use. Lyophilized powder was dissolved in 0.25% carboxymethyl cellulose before use in animal experiments.

### 4.3. Animals

Male 7-week-old Sprague–Dawley rats were purchased from Samtako Inc. (Osan, Gyeonggi-do, Korea) and housed under controlled conditions with a 12 h light/dark cycle. They were maintained for 1 week prior to experiments. The rats were provided with a laboratory diet and water ad libitum. All animal experimental protocols used in this study were approved by the Institutional Animal Care and Use Committee of Daejeon University (DJU-IACUC-2017-032). The MIA-induced OA rat model (Merck KGaA, Darmstadt, Germany) was established as described previously [27,36]. To induce OA, the rats were directly injected with MIA (3 mg in 50 μL of 0.9% saline) in the intra-articular space of the right knee under anesthetic with ether. All rats were divided randomly into five groups (*n* = 7 per group) as follows: (1) normal group (NC, saline treatment and no MIA injection); (2) MIA-induced OA group (MIA, saline treatment, and MIA injection); (3) DAE-treated group (MIA + DAE, 200 mg/kg of DAE treatment and MIA injection); (4) PUE-treated group (MIA + PUE, 200 mg/kg of PUE treatment and MIA injection); (5) IM-treated group (MIA + IM, 1 mg/kg of IM treatment and MIA injection). Extracts were administered by oral gavage once a day for 3 weeks. A schematic representation of the experimental protocol is provided in Figure 6B.

### 4.4. Measurement of Hind Paw Weight-Bearing Distribution

In the MIA-induced OA model, changes in weight-bearing distribution are used as a measure of disease progression and efficacy of anti-inflammatory compounds, and as a joint discomfort index [27]. After OA induction, weight-bearing was measured once a week using an incapacitance tester (Linton Instrumentation, Norfolk, UK). The rats were carefully placed in the measurement chamber, and the weight-bearing force exerted by the hind limb was measured and averaged over a period of time. The weight percentage distributed over the treated (ipsilateral) hind limb was calculated as follows [37]: [Weight on ipsilateral hind limb = (Weight on ipsilateral + Weight on contralateral)] × 100(1)

### 4.5. Biochemical Blood Analysis

At sacrifice, blood samples were centrifuged at 1500× *g* for 15 min, and serum was stored at −70 °C until analysis. Serum AST and ALT activity was measured using a Hitachi 7080 automatic analyzer (Hitachi Co., Tokyo, Japan). Serum levels of TNF-α and IL-1β were measured using ELISA kits from R&D Systems (Minneapolis, MN, USA) according to the manufacturer’s instructions.

### 4.6. Histopathological Analysis

Following rat sacrifice, tissue specimens were removed from the knee joint, fixed in 10% formalin, embedded in paraffin, and serially sectioned. H&E or Safranin O staining was performed to visualize joint cells and matrices. Histological changes were examined by light microscopy using an Olympus CX31/BX51 instrument (Olympus Optical Co., Tokyo, Japan) and photographed with an Olympus DP70 camera.

### 4.7. Immunohistochemical Analysis

For immunohistochemistry, the tissue sections were deparaffinized and subjected to antigen repair and blocking as described previously [38]. Following incubation overnight at 4 °C with primary antibodies against MMP-2 (1:100; #40994; Cell Signaling Technology, Danvers, MA, USA), MMP-9 (1:100; #13667s; Cell Signaling Technology), and MMP-13 (1:100; ab39012; Abcam, Cambridge, UK), the samples were exposed to horseradish peroxidase-conjugated anti-rabbit IgG secondary antibody (VECTASTAIN Elite ABC Kit; Vector Laboratories, Burlingame, CA, USA). Subsequently, the peroxidase reaction was developed using diaminobenzidine substrate (DAB Kit, SK-4100; Vector Laboratories). All sections were counterstained with Harris’s hematoxylin prior to mounting. Stained specimens were observed using a microscope BX51 (Olympus, Tokyo, Japan) equipped with a DP70 digital camera (Olympus).

### 4.8. Statistical Analysis

Statistical analyses were performed using GraphPad Prism Software v.7.0 for Windows (GraphPad Software, La Jolla, CA, USA). Differences between the five groups were analyzed by one-way analysis of variance (ANOVA) with Dunnett’s multiple comparisons tests or two-way ANOVA with Tukey’s multiple comparisons tests. The results were considered statistically significant if two-tailed *p*-values were <0.05.

## 5. Conclusions

DAE and PUE showed potent effects in an MIA-induced OA model. DAE- and PUE-treated rats recovered hind paw weight-bearing ability compared with MIA-induced rats, and both agents restored balance and relieved joint pain, although PUE was more effective than DAE. The results suggest that *P. umbrosa* could be used as an alternative herb for *D. asperoides* for the treatment of OA.

## Figures and Tables

**Figure 1 plants-10-02030-f001:**
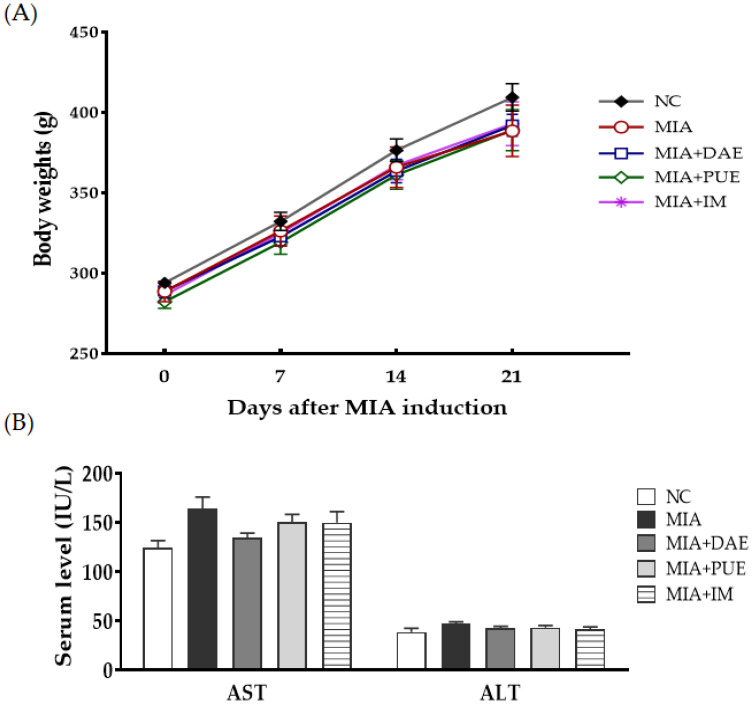
Effects of *Dipsacus asperoides* extract (DAE) and *Phlomis umbrosa* extract (PUE) on body weight and serum aminotransferase levels in MIA-induced OA rats. (**A**) The body weight of rats was measured once per week for 3 weeks. (**B**) Serum aspartate aminotransferase (AST) and alanine aminotransferase (ALT) levels. Data are expressed as the mean ± SEM (*n* = 7 per group). NC, untreated; MIA, only MIA-induced; MIA + DAE, MIA-induced and DAE-treated; MIA + PUE, MIA-induced and PUE-treated; MIA + IM, MIA-induced and Indomethacin (IM)-treated rats.

**Figure 2 plants-10-02030-f002:**
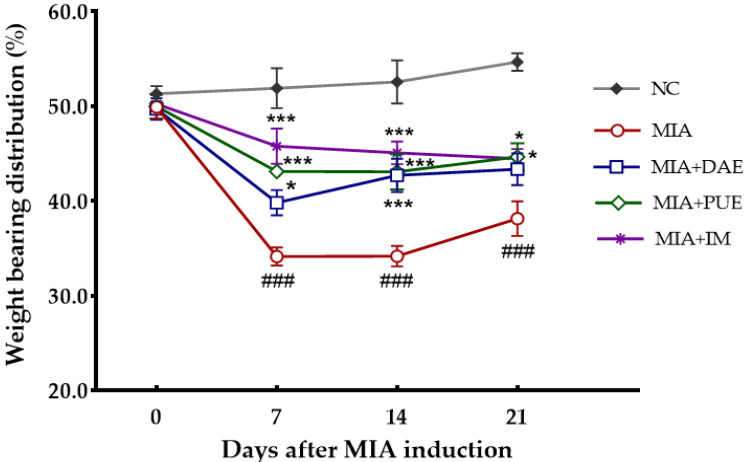
Effects of DAE and PUE on hind paw weight-bearing distribution in MIA-induced OA rats. After the injection of MIA, weight-bearing distribution was measured using an incapacitance tester once per week for 21 days. Data are expressed as the mean ± SEM (*n* = 7 per group). NC, untreated; MIA, only MIA-induced; MIA + DAE, MIA-induced and DAE-treated; MIA + PUE, MIA-induced and PUE-treated; MIA + IM, MIA-induced and IM-treated rats. ### *p* < 0.001 indicates statistically significant differences between the NC and MIA-induced control rats. ** p* < 0.05, **** p* < 0.001 indicate statistically significant differences between the MIA-control rats and the DAE-, PUE-, or IM-treated rats.

**Figure 3 plants-10-02030-f003:**
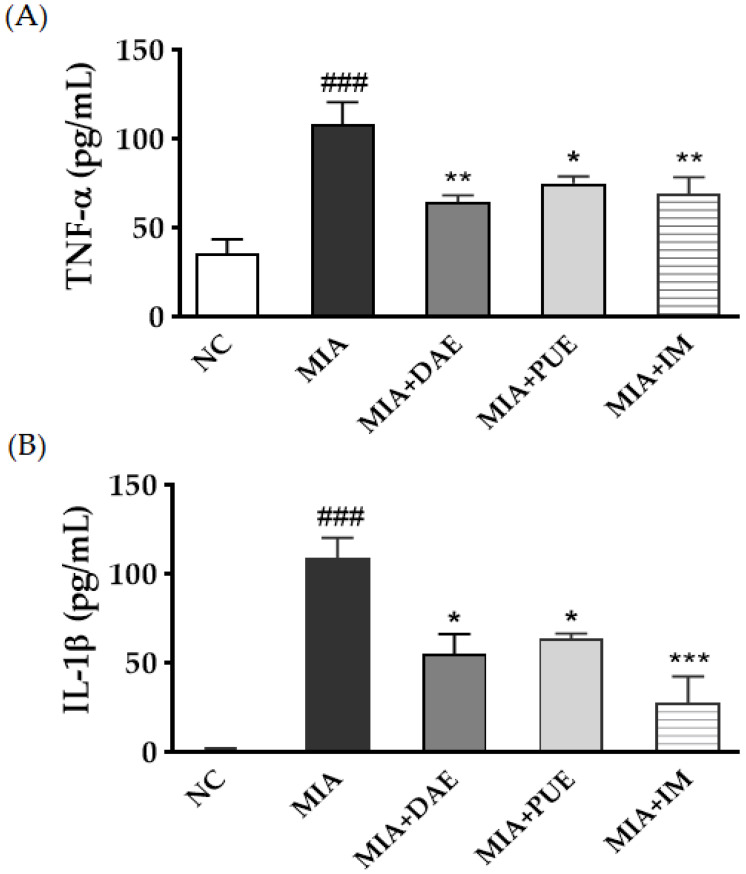
Effects of DAE and PUE on the serum levels of cytokines in MIA-induced OA rats. Serum cytokines including (**A**) TNF-α and (**B**) IL-1β were quantified by ELISA. Data are expressed as the mean ± SEM (*n* ≥ 3). NC, untreated; MIA, only MIA-induced; MIA + DAE, MIA-induced and DAE-treated; MIA + PUE, MIA-induced and PUE-treated; MIA + IM, MIA- induced and IM-treated rats. *### p* < 0.001 indicates statistically significant differences between the NC and MIA-induced control rats. ** p* < 0.05, *** p* < 0.01, and **** p* < 0.001 indicate statistically significant differences between the MIA-induced control rats and the DAE-, PUE-, or IM-treated rats.

**Figure 4 plants-10-02030-f004:**
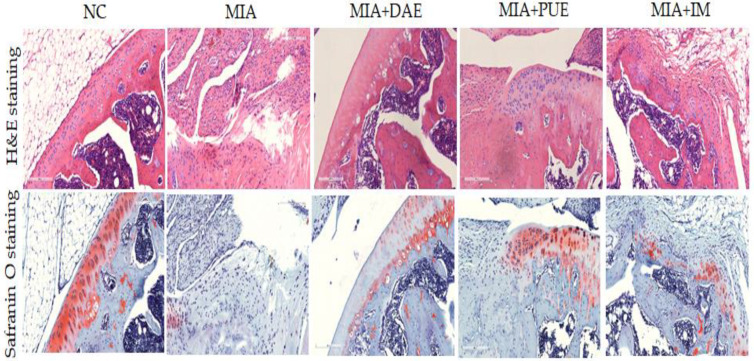
Effects of DAE and PUE on histopathological features of knee joint tissue in MIA-induced OA rats. Representative photographs of knee joint sections stained with H&E and Safranin O (×100 magnification). NC, untreated; MIA, only MIA-induced; MIA + DAE, MIA-induced and DAE-treated; MIA + PUE, MIA-induced and PUE-treated; MIA + IM, MIA-induced and IM-treated rats.

**Figure 5 plants-10-02030-f005:**
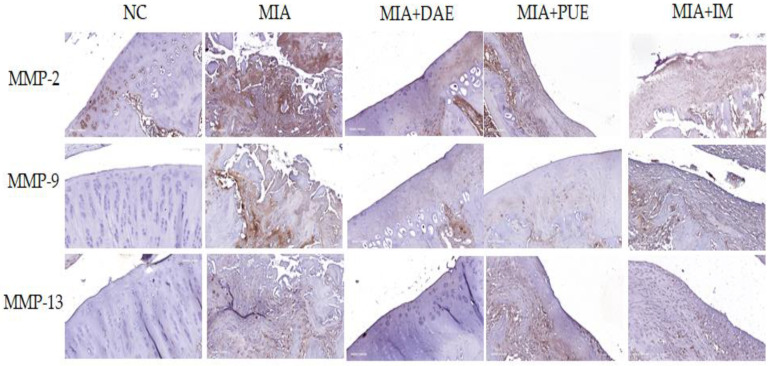
Effects of DAE and PUE on the expression of MMP-2, MMP-9, and MMP-13 in knee joint tissues of MIA-induced OA rats. Representative images of immunohistochemical staining of MMP-2, MMP-9, and MMP-13 expression in knee joint tissues. NC, untreated; MIA, only MIA-induced; MIA + DAE, MIA-induced and DAE-treated; MIA + PUE, MIA-induced and PUE-treated; MIA + IM, MIA-induced and IM-treated rats.

**Figure 6 plants-10-02030-f006:**
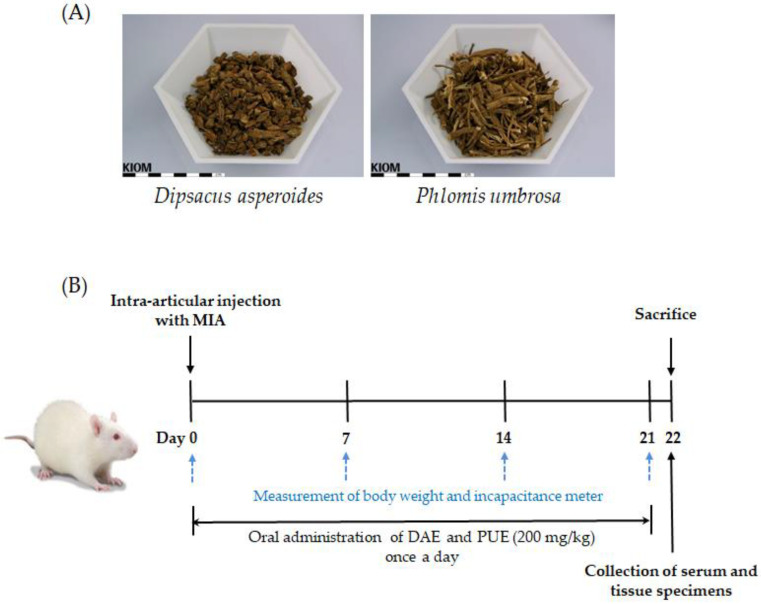
Experimental plant materials and protocol. (**A**) Representative photographs of dried roots of *Dipsacus asperoides* and *Phlomis umbrosa* used in this study. (**B**) The experimental protocol for inducing osteoarthritis (OA) and treatment with extracts. SD rats were divided randomly into five groups: NC, MIA, MIA + DAE, MIA + PUE, and MIA + IM (*n* = 7 per group). MIA, monosodium iodoacetate; DAE, *Dipsacus asperoides* extract; PUE, *Phlomis umbrosa* extract; IM, indomethacin.

## Data Availability

Not applicable.

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
