# Peer review of "Effects of Dipsacus asperoides and Phlomis umbrosa Extracts in a Rat Model of Osteoarthritis"

_plants, 2021, doi:10.3390/plants10102030_

Round 1

Reviewer 1 Report

The authors studied the effect Dipsacus asperoides (DAE) and Phlomis umbrosa (PUE) extract on osteoartritis (OA) in a monosodium iodoacetate induced OA rat model. They demonstrated the efficacy of both extracts, which is a valuable contribution to the use of these herbal products to the traditional Korean medicine.

This year, the authors published two papers on the bioactivity of DAE and PUE on monosodium iodoacetate induced OA rat model. This manuscript is the third member of that series.

The manuscript is suitable for publication after minor revision.

Further information on the preparation and the components of the plant materials (PUE and DAE) would be useful.

In lines 117-123, there is a typographical error, there is no margin and the lines are centered middle.

Line 358.: monosodium iodoacetate.

A careful check of the use of capital letters, italic, etc. in the list of references would be necessary.

Author Response

7th September, 2021

Plants

Ref. No.: plants-1374175

Dear Editor,

We appreciate the opportunity to respond to our reviewer’s comments. Our manuscript is now revised to incorporate reviewer’s meaningful critiques. Major changes in the text are highlighted in red in the revised manuscript. In addition, please find below a point-by-point response to the reviewers’ comments. Please see the attachment.

Reviewer 2 Report

The introduction part should be improved. A ethnopharmacological review of both species could be included and, overall, it is necessary to justify why the authors have selected those; the statement provided is not enought and at least should be supported by bibliography (third paragraph). 

"Before in vivo experiments, D. asperoides and P. umbrosa were authenticated using morphological and genetic characterization, as well as phytochemical analysis for definitive authentication and quality control as described previously"; it should be explained briefly.  

"The origin of these herbs was analyzed previously, and the two species were authenticated using DNA barcodes, the chloroplast genome, and sequence-characterized amplified region (SCAR) markers [9]. The origin of the sample used in the present study was determined previously, and the sample was characterized by quantitative analysis [16,17]." Please indicate the importance of those findings or why it is relevant here. 

I found specially interesting the paragraph about the disponibility of both species. 

Author Response

(The authors gave the same response as above.)
